# Transmission of SARS-CoV-2 from humans to animals and potential host adaptation

Cedric C. S. Tan [1,2 ✉], Su Datt Lam[3,4], Damien Richard[1,5], Christopher J. Owen [1], Dorothea Berchtold[1], Christine Orengo [4], Meera Surendran Nair[6,7], Suresh V. Kuchipudi [6,7], Vivek Kapur[7,8], Lucy van Dorp [1,9] & François Balloux [1,9]

SARS-CoV-2, the causative agent of the COVID-19 pandemic, can infect a wide range of mammals. Since its spread in humans, secondary host jumps of SARS-CoV-2 from humans to multiple domestic and wild populations of mammals have been documented. Understanding the extent of adaptation to these animal hosts is critical for assessing the threat that the spillback of animal-adapted SARS-CoV-2 into humans poses. We compare the genomic landscapes of SARS-CoV-2 isolated from animal species to that in humans, profiling the mutational biases indicative of potentially different selective pressures in animals. We focus on viral genomes isolated from mink (*Neovison vison*) and white-tailed deer (*Odocoileus virginianus*) for which multiple independent outbreaks driven by onward animal-to-animal transmission have been reported. We identify five candidate mutations for animal-specific adaptation in mink (NSP9_G37E, Spike_F486L, Spike_N501T, Spike_Y453F, ORF3a_L219V), and one in deer (NSP3a_L1035F), though they appear to confer a minimal advantage for human-to-human transmission. No considerable changes to the mutation rate or evolutionary trajectory of SARS-CoV-2 has resulted from circulation in mink and deer thus far. Our findings suggest that minimal adaptation was required for onward transmission in mink and deer following human-to-animal spillover, highlighting the 'generalist' nature of SARS-CoV-2 as a mammalian pathogen.

[1] UCL Genetics Institute, University College London, London, UK. [2] Genome Institute of Singapore, A*STAR, Singapore, Singapore. [3] Department of Applied Physics, Faculty of Science and Technology, Universiti Kebangsaan Malaysia, Bangi, Selangor, Malaysia. [4] Department of Structural and Molecular Biology, University College London, London, UK. [5] Division of Infection and Immunity, University College London, London, UK. [6] Animal Diagnostic Laboratory, Department of Veterinary and Biomedical Sciences, The Pennsylvania State University, PA Pennsylvania, USA. [7] Huck Institutes of the Life Sciences, The Pennsylvania State University, PA Pennsylvania, USA. [8] Department of Animal Science, The Pennsylvania State University, PA Pennsylvania, USA. [9] These authors jointly supervised this work: Lucy van Dorp, François Balloux. ✉email: cedriccstan@gmail.com

Severe acute respiratory syndrome coronavirus 2 (SARS-CoV-2) is able to infect multiple mammalian host species[1], which is a characteristic seen in other coronaviruses[2]. SARS-CoV-2 emerged in humans in late 2019[3], presumably after animal-to-human transmission (i.e., zoonotic spillover) of an ancestral viral lineage belonging to the subgenus *Sarbecovirus* that circulated in bats[4,5], although its proximal origin remains unresolved. The evolutionary events after the zoonotic host jump but preceding the initial COVID-19 outbreak in Wuhan, China (late December 2019) remain poorly characterised due to the lack of genomic sequences collected during that period. Further, SARS-CoV-2 has likely been circulating in humans for some time before it was formally detected. This is expected given the similarity of disease presentation relative to other respiratory infections and the high rate of asymptomatic infections in humans (~40%)[6]. Indeed, a critical mass of cases presenting with severe disease must be reached before alerting infectious disease surveillance efforts.

For a successful host jump of a pathogen and its subsequent emergence, several traits must be acquired. One key prerequisite is the ability to infect cells of the novel host, which depends on the presence of compatible host cell receptors. SARS-CoV-2 can infect cells of multiple mammalian host species[7–9], primarily due to the conservation of the angiotensin-converting enzyme 2 (ACE2), the primary host cell receptor used for viral entry, across mammals[10–12]. Another essential trait is the ability to transmit efficiently within the populations of the novel host. Infections of host populations that do not efficiently transmit the pathogen further, also known as 'dead-end' hosts, may quickly lead to pathogen extinction within that population. Dogs, which are susceptible to SARS-CoV-2 infection but do not efficiently transmit the virus[1] are a possible example of a dead-end host. On the other hand, human-to-human transmission is rapid, with early estimates of the mean number of subsequent infections produced by an infectious person in a totally naïve population (i.e., basic reproductive number, $R_0$) ranging from 1.5–6.5[13].

Evolutionary analyses of SARS-CoV-2 and close relatives suggest that both efficient human-to-human transmission and ACE2 usage were not acquired recently, but may already have been present in ancestral bat-associated lineages[4,5]. While these findings have not been demonstrated experimentally, this suggests that SARS-CoV-2 could have been well pre-adapted for circulation in humans prior to its emergence. Consistent with this, early efforts to identify mutations associated to the transmissibility of SARS-CoV-2 failed to identify obvious candidates for adaptation to its human host[5,14]. However, with the recent emergence of more transmissible Variants of Concern (VoC) such as Alpha, Delta and Omicron that have higher $R_0$ values[15–17], it is generally accepted that SARS-CoV-2 is still adapting to its human host[18], maintaining its fitness in the face of increasing vaccine coverage and infection-acquired immunity in the human population.

After its initial zoonotic host jump into humans, multiple secondary host jumps of SARS-CoV-2 from humans into animals (i.e., human-to-animal spillover) and significant transmission have been reported for domestic and wild mammals. This offers potential insights into the early evolutionary dynamics leading to and following host jumps. As of 17th March 2022, a total of 1282 high quality SARS-CoV-2 genomes associated with natural or experimental infection of 25 animal species have been deposited on GISAID[19,20] (Table 1). The first animal-associated outbreaks seeded by human-to-animal spillover events emerged in mink farms in the Netherlands in April 2020[21], and subsequently in Denmark in June 2020[22,23], where transmission was rapid. Indeed, initial testing found that 65% of mink (*Neovison vison*) in

Danish mink farms had been infected by late June 2020[24]. Further, SARS-CoV-2 in minks were found to transmit readily back into humans (i.e., spillback)[25]. These findings prompted culls of minks in Dutch mink farms in early June 2020[26]. Separately, in November 2020, an initial report from Denmark raised concerns about the emergence of a mink-associated SARS-CoV-2 lineage circulating in humans and minks of farms in Northern Jutland, Denmark[27,28]. This 'mink-derived' lineage, known as the 'cluster 5 variant', possessed five mutations in the Spike protein (H69/V70 deletion, Y453F, D614G, I692V, M1229I) and showed some evidence of partial immune escape[27,29,30], which led to the subsequent decision to cull approximately 17 million Danish minks[31].

In the last quarter of 2021, studies reporting human-to-animal spillover into wild white-tailed deer (*Odocoileus virginianus*) in the USA began to surface[32–35]. White-tailed deer are one of the most abundant wild ruminants in the USA, and some of these spillover events were associated with the start of the regular deer hunting season[33]. Significant onward transmission was observed, with ~30% of sampled deer being SARS-CoV-2-positive in Iowa[33] and Ohio[32], and a reported 40% seroprevalence across four US states[35].

Of fundamental interest is whether SARS-CoV-2 required host-adaptive mutations to jump into animal hosts, the extent of host-specific adaptation following its host jumps, and how the introduction of SARS-CoV-2 into animals impacts the evolutionary trajectory of the virus. Given the rapid and extensive onward transmission in mink and deer, there was likely ample opportunity for the virus to adapt to circulation in these host populations. Further, the rapid testing and intensive sequencing efforts early into these outbreaks offer a glimpse into the key evolutionary events surrounding spillovers and the establishment of new host reservoirs.

In this work, we focus on published and publicly available sequences isolated from mink and deer, analysing these animal-associated sequences relative to carefully curated subsamples of human SARS-CoV-2. In particular, we look for changes in mutational biases, genomic composition, and mutation rates in animal SARS-CoV-2 clusters relative to human-associated counterparts. Additionally, we screen for mutations that may have arisen due to host-specific adaptation and subsequently assess the potential impact of these mutations bioinformatically. We find that circulation of SARS-CoV-2 in mink and deer has resulted in some degree of viral adaptation to its animal host but not in elevated mutation rates nor in significant changes to the evolutionary landscape of the virus. Our findings suggest that efficient animal-to-animal transmission of SAR-CoV-2 in these hosts required minimal adaptation, highlighting the 'generalist' nature of the virus.

## Results

**Multiple human-to-animal spillover events of SARS-CoV-2.** Following the global spread of SARS-CoV-2, spillover of the virus from humans into domestic and wild animal species have been documented. Placement of the animal-associated genomes shown in Table 1 recapitulates these multiple independent human-to-animal spillover events (Fig. 1a). The clustering of animal isolates on the global phylogeny correlates well with the different species-specific transmission potentials and the extent of transmission amongst animal populations, though this could also in part be due to differential sampling efforts. Cat and dog isolates appear as highly polyphyletic singletons reflecting the poor animal-to-animal transmission in companion animals in addition to sparse sequencing efforts. Separately, we find small clusters of isolates from big cats (i.e., *Panthera* spp.), reflecting outbreaks of SARS-

**Table 1 Summary of high-quality animal-associated SARS-CoV-2 genomes.**

| Host taxon | Common name | Population type | No. of isolates | No. of countries | Onward transmission |
|---|---|---|---|---|---|
| *Neovison vison* | American mink | Farmed | 928 | 9 | Yes[21–23,25,85,86] |
| *Odocoileus virginianus* | White-tailed deer | Wild | 95 | 1 | Yes[32–35,87] |
| *Felis catus domesticus* | Domestic cat | Pet/stray | 76 | 13 | Yes[88–91] |
| *Canis lupus familiaris* | Domestic dog | Pet/stray | 40 | 6 | Unlikely[92] |
| *Panthera* spp. | Big cats | Captive | 87 | 6 | Yes[36,38] |
| *Mesocricetus auratus* | Golden hamster | Pet | 20 | 3 | Yes[93] |

Animal taxa that were associated with <10 isolates were excluded from this table for brevity.

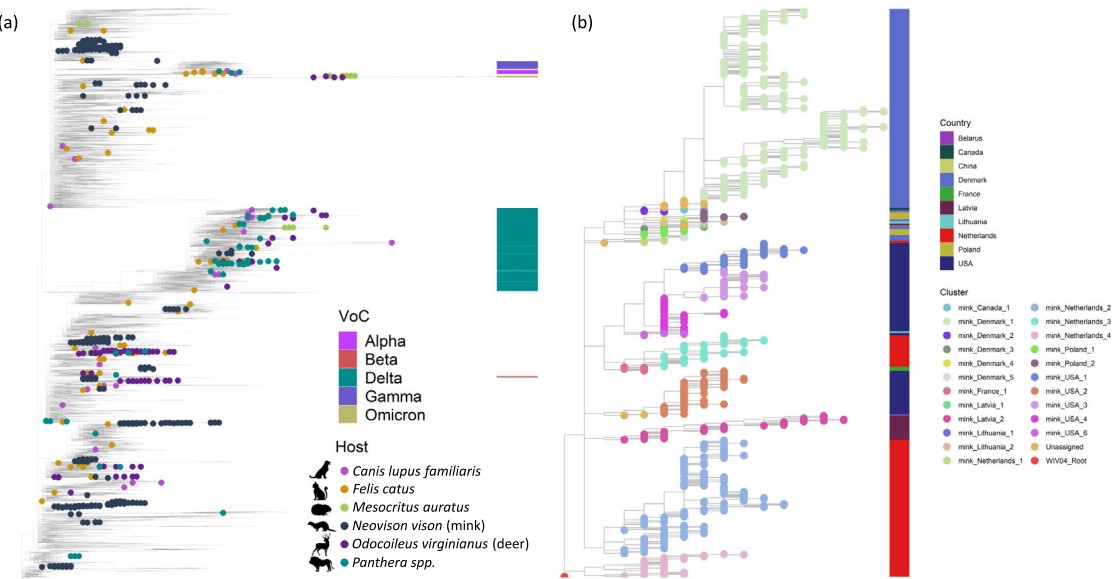

**Fig. 1 Multiple emergences and onward transmission of SARS-CoV-2 in animals. a** Subsampled Audacity tree ($n = 16,911$) comprising 10 human isolates per PANGO lineage, and all animal isolates shown in Table 1, illustrating the global context of SARS-CoV-2 infections in animals. **b** Maximum-likelihood tree of all 928 mink isolates, with manually curated cluster names (see "Methods") and country of isolation annotated.

CoV-2 in multiple species of captive zoo animals around the world[36–39].

Manual inspection of the global phylogeny supports a minimum number of 22 and seven well-supported, phylogenetically distinct clusters of SARS-CoV-2 in mink and deer due to independent spillover events of multiple human SARS-CoV-2 lineages. Several large clusters were observed in mink (Fig. 1b), with the largest mink cluster in Denmark[22] reaching >300 sequenced infections. This reflects the efficient mink-to-mink transmission of SARS-CoV-2 in intensive farming settings. Additionally, we find multiple moderately sized clusters of SARS-CoV-2 in deer that represent frequent spillover events due to the geographical overlap of deer and human habitats, followed by substantial deer-to-deer transmission.

Finally, we find that the animal outbreaks were seeded by 89 of the 1,591 PANGO lineages[40] that have been defined as circulating in humans prior to 17 March 2022, including the Alpha, Delta, Omicron, Iota, Epsilon and Mu variants. The 89 PANGO lineages found in animals are not restricted to particular clades of the global diversity of SARS-CoV-2 and instead appear to be broadly representative of the different lineages circulating in humans. This suggests that efficient onward transmission to animals is not a property of any particular subset of SARS-CoV-2 lineages in circulation in humans.

**Homoplasy and allele frequency analyses identify candidate mutations for host-specific adaptation.** To identify candidates

for host adaptation, we compared mink SARS-CoV-2 sequences to a roughly similar number of human isolates with matching PANGO lineage, range of sampling dates and country origin (human background 1, see "Methods"). This allowed us to identify 20 and 34 candidate mutations which may be the result of mink or deer-specific adaptation, respectively. These mutations were (A) at a two-fold higher allele frequency in animal than human isolates and (B) had an animal allele frequency > 0.1, or (C) have emerged at least thrice independently in each animal host-only phylogeny (Fig. 2a, b). Since spillover events involve only a subset of human viral lineages, selectively neutral mutations that were already present in these lineages may appear homoplastic following spillover into independent animal populations. As such mutations that (D) were not inherited from the parent human lineage are more likely to be adaptive. This can be determined by visually inspecting the animal isolates in the context of human background 1 (Fig. S1a). The genomic and residue positions, allele frequencies and the number of emergences for the 20 putative mink- and 34 deer-specific candidate mutations are shown in Supplementary Data 1.

Of the identified mutations, four non-synonymous changes in minks (NSP9_G37E, Spike_F486L, Spike_N501T, ORF3a_L219V) and one in deer (NSP3_L1035F) fulfilled all four criteria in addition to (E) being present in at least three independent clusters (Fig. 2c, d), are the strongest candidates for putative host adaptation. Three synonymous changes satisfying criteria (A)–(E) (NSP6_C11572T, NSP3a_C7303T, NSP4_C9430T) were also found in deer-associated

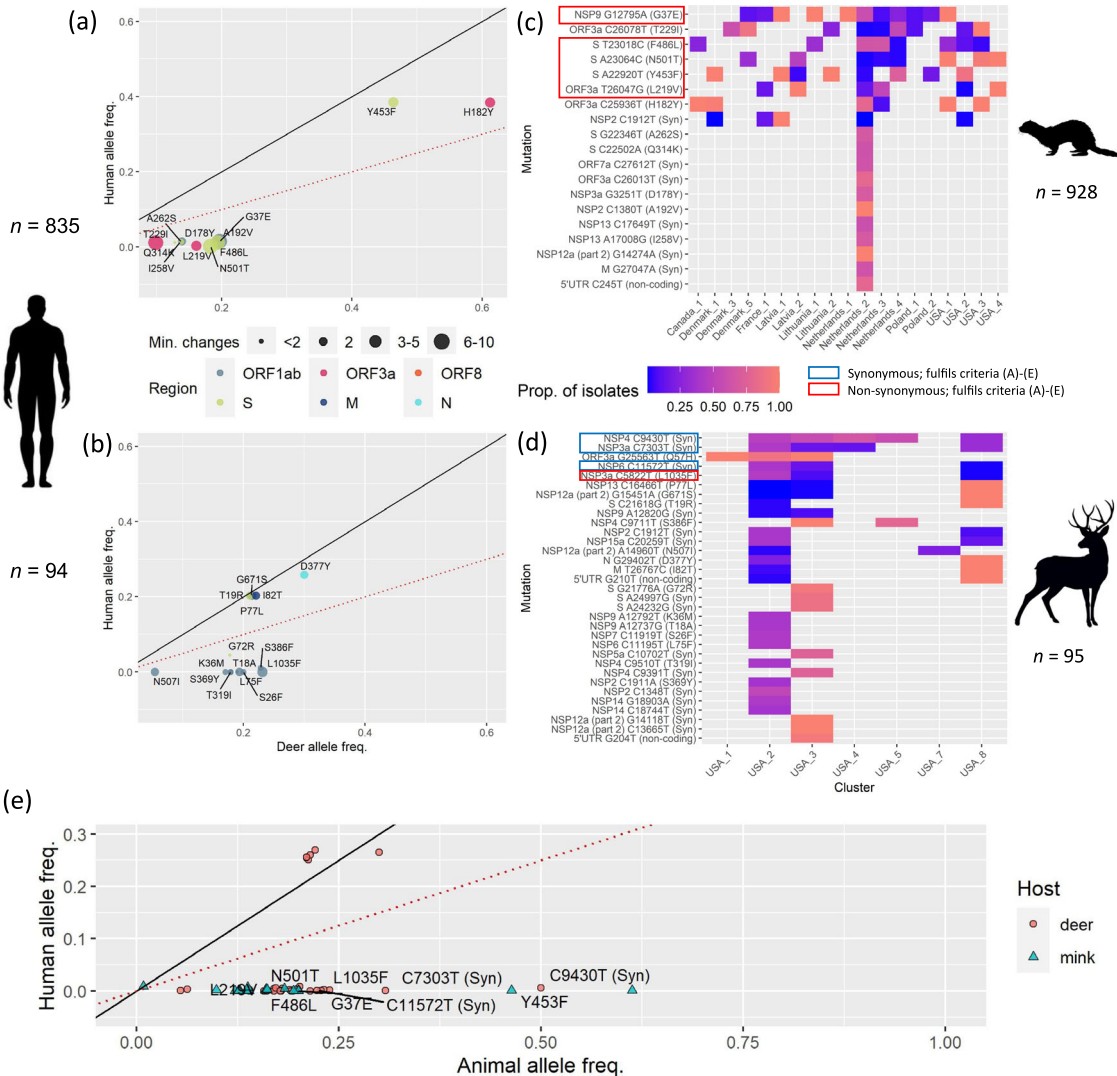

**Fig. 2 Homoplasy and allele frequency analysis.** Scatter plot of putatively adaptive non-synonymous mutations in **a** mink and **b** deer. Point size represents the minimum number of independent emergences for each mutation in a phylogeny reconstructed from 928 mink or 95 deer isolates. Human isolates with matching PANGO lineages, from the same countries, and that were sampled within the range of sampling dates of mink ($n = 835$) or deer isolates ($n = 94$), were used to compute the human background allele frequencies (human background 1). The dotted red lines and solid black lines, indicate where the allele frequencies in each animal host are two-fold that in humans, and where the human and animal allele frequencies are equal, respectively. Heatmap visualising the proportions of mutation-carrying SARS-CoV-2 isolates within manually curated phylogenetic clusters in **c** mink and **d** deer. **e** Allele frequencies of 20 mink and 34 deer candidate mutations in human background 2. The strongest candidate non-synonymous and synonymous mutations satisfying criteria (A)–(E) are indicated by red and blue boxes, respectively. The genomic region associated with each mutation is given by the colour in panel **b**.

SARS-CoV-2 but may have a more cryptic relationship with protein function and host adaptation. Interestingly, none of the strong candidate mutations satisfying criteria (A)–(E) in deer were found in the Spike protein.

Notably, though Spike_Y453F has been shown to improve Spike:mink-ACE2 interactions and suggested to be mink-adaptive[29,30], its frequency in mink-associated virus is comparable to those considered in human background 1. Inspection of a subsampled phylogeny comprising all mink and human isolates collected in Denmark prior to 1 December 2020 (Fig. S1b) found that the mink and human isolates in the mink_Denmark_1 cluster are interspersed, suggesting complex back-and-forth transmission patterns between minks and humans. This makes it difficult to interpret whether the mutation first arose in human lineages and spilled over into minks, or the inverse. Nevertheless, excluding the mink_Denmark_1 cluster results in Y453F

occurring at greater than two-fold frequency in minks relative to humans, satisfying criterion (A). We therefore consider Y453F to also be a strong candidate mink-adaptive mutation.

Separately, we did not find any mutations that were fixed in the animal populations and at a considerably lower frequency in humans. Under a scenario where key host-specific mutations must be acquired for an expansion of host tropism and subsequent spillover, we expect such mutations to be fixed in viruses isolated from the novel animal host, but at a lower frequency in the primary host. The absence of fixed mutations suggests that host-specific adaptation was not necessary for human-to-animal spillover of SARS-CoV-2 into mink and deer.

Finally, we compared the frequencies of candidate mutations in animals relative to those in all human lineages within the same country regardless of sampling time (human background 2) to infer host-specific selective pressures acting on these mutations.

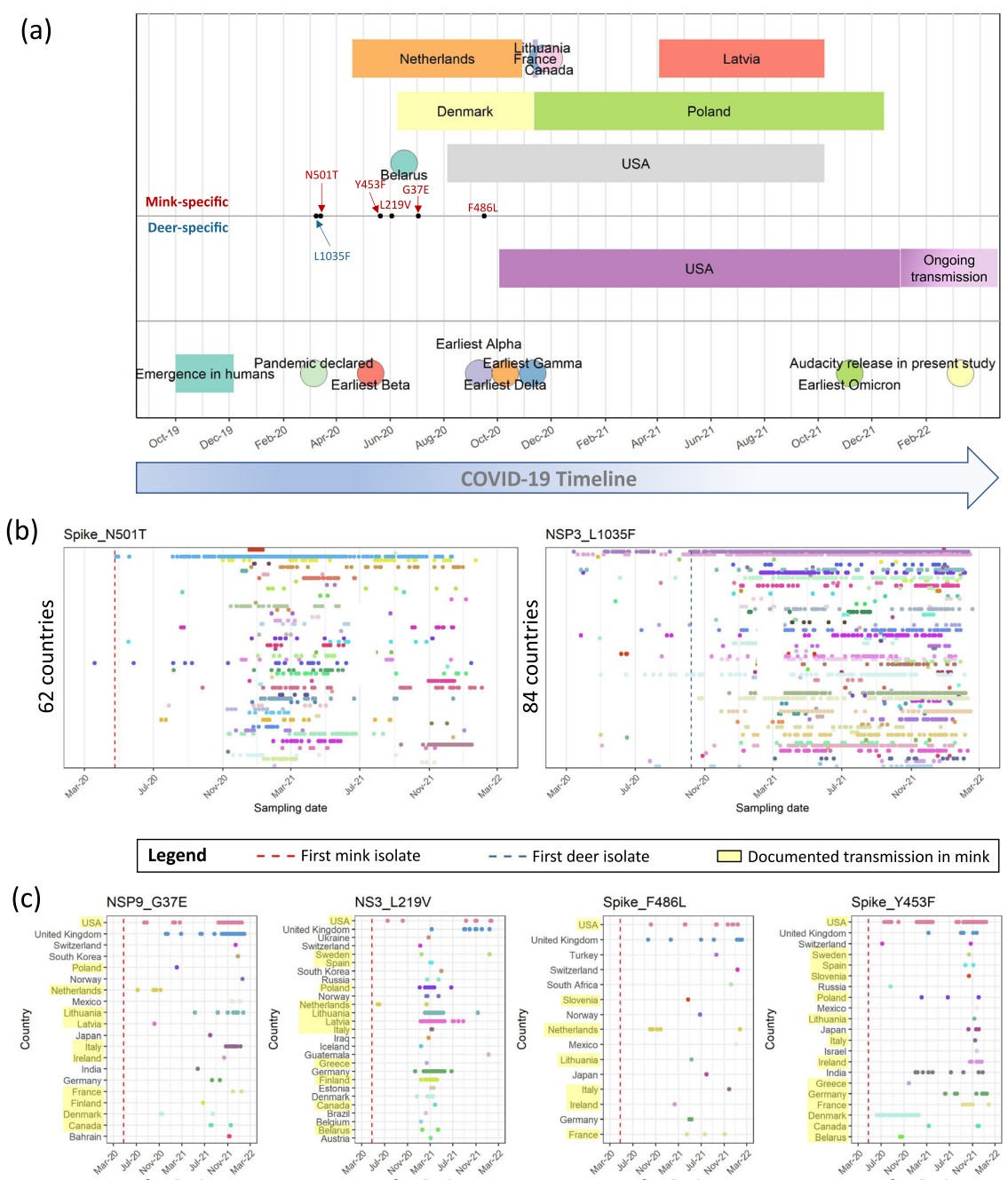

**Fig. 3 Timeline of the COVID-19 pandemic. a** The key events of the pandemic from the estimated emergence of SARS-CoV-2 in humans[3] to the sampling dates of the first isolates for each VoC are annotated in the lowest panel. The coloured rectangles in the upper first and second panels indicate the range of sampling dates of animal-associated SARS-CoV-2 sequences in the different countries. The sampling dates of the earliest human isolates carrying each candidate mutation are annotated along the timeline are indicated by black points. Panels **b** and **c** show the temporal distributions of candidate mutations in human SARS-CoV-2 isolates collected prior to 17th March 2022. Red and blue dashed lines indicate the sampling date of the first mink-associated isolate in the Netherlands and deer-associated isolate in the USA, respectively. For panel **b**, country names were omitted, and the number of countries where the candidate mutations were found in human isolates are annotated. For panel **c**, countries where human-to-mink transmission has been documented are highlighted in yellow[41].

We find that all of the strongest candidate mutations prevalent in animal isolates are almost non-existent in human isolates (Fig. 2e), suggesting that while these mutations may be tolerated/adaptive in animals, they may be selected against in humans.

**Emergence of some candidate animal-adaptive mutations predates documented human-to-animal spillovers.** We placed the range of sampling dates of animal isolates in the context of the

broader COVID-19 pandemic timeline (Fig. 3a). Two of the six strongest non-synonymous candidates (mink: Spike_N501T, deer: NSP3_L1035F) emerged in humans early in the initial wave of the pandemic, predating the first documented SARS-CoV-2 outbreaks in their respective animal hosts. Further, even before the detection of SARS-CoV-2 in deer, NSP3_L1035F had already emerged in 20 other countries excluding the USA (Fig. 3b). Spike_N501T and NSP3_L1035F were also found in human isolates distributed across

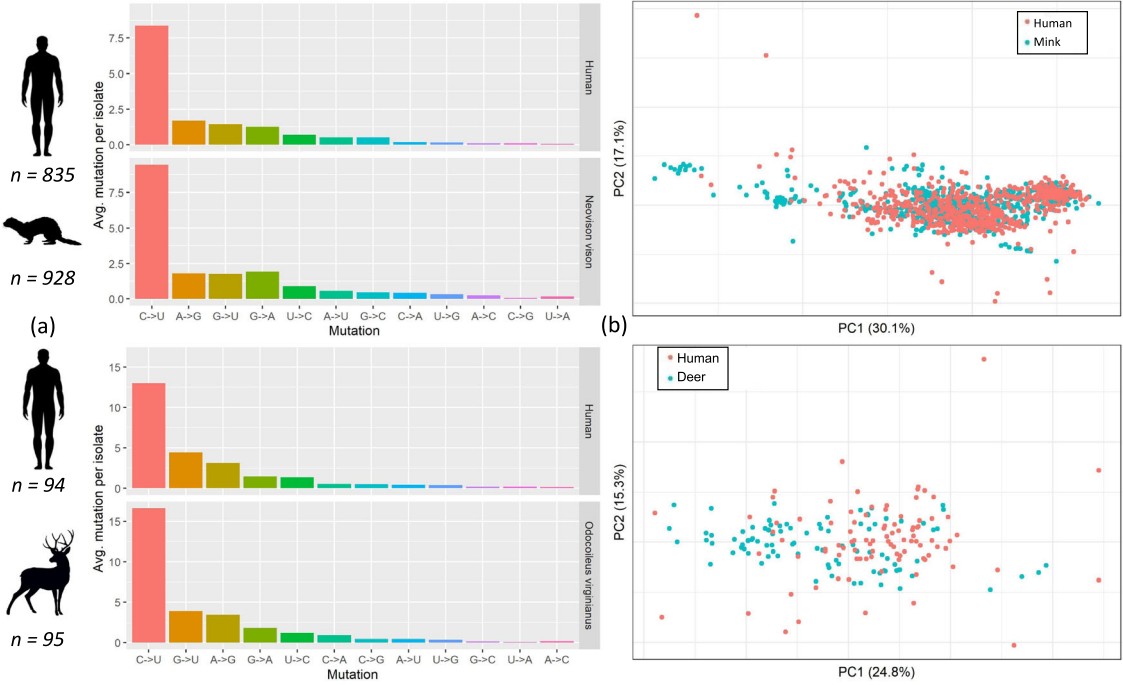

**Fig. 4 Host-specific genomic landscapes. a** Nucleotide-nucleotide transition frequencies (x-axis) against average mutations observed per isolate in human and animal hosts (as indicated by symbols), and **b** principal components analysis of all dinucleotide frequencies, stratified by host.

a large timespan and across 62 and 84 countries, respectively, even where mink or deer populations are not present (Fig. 3b). Since these mutations are not associated with any highly transmissible VoCs and that we find no evidence of strong positive selection acting on these mutations in humans (Fig. 2e; see the previous section), carriage of these alleles across many countries is not expected. These observations support the hypothesis that the two mutations may have emerged during the evolutionary history of SARS-CoV-2 in humans independent of human-to-animal-to-human transmission. On the other hand, the four strongest candidate mink-adaptive mutations (NSP9_G37E, ORF3a_L219V, Spike_F486L, Spike_Y453F) emerged in humans after the first mink outbreaks in the Netherlands. Some of the earliest human isolates carrying these mutations were first sequenced in the Netherlands and Denmark. Further, the human isolates carrying these mutations tend to originate from countries where human-to-mink SARS-CoV-2 transmission has been reported[41] (Fig. 3c). These findings suggest an association of these mutations with human-to-mink spillover and subsequent spillback.

**Immediate changes to genomic composition in animal isolates.** To investigate changes to the genomic landscape of SARS-CoV-2 immediately following a host jump, we analysed the nucleotide-nucleotide transitions and dinucleotide frequencies of animal isolates relative to human background 1. The proportions of nucleotide-nucleotide transitions differed between mink ($\chi^2 = 245.3$, $p < 0.001$) and deer ($\chi^2 = 37.5$, $p < 0.001$) relative to those in human isolates (Fig. 4a). However, the overall mutational profiles are similar with $C \rightarrow U$ transitions dominating. Consistently, a principal components analysis of dinucleotide frequencies shows highly overlapping host clusters, indicating that the genome composition of SARS-CoV-2 infecting different hosts does not differ considerably (Fig. 4b). Of note, $A \rightarrow G$ transitions appear to occur less frequently in mink than humans (permutation test, $p < 0.001$), though this change is subtle compared to the overrepresentation of $C \rightarrow U$ mutations (Fig. 4a). Direct comparisons between mink and deer, or

between the two human backgrounds could not be made due to the imbalanced representation of PANGO lineages.

**Spillovers into novel animal hosts did not lead to inflated substitution rates.** We attempted to tip-calibrate animal-human maximum-likelihood phylogenies, comprising either mink or deer isolates with their corresponding human backgrounds (background 1). Root-to-tip regressions for isolates from each country suggest that only mink isolates from Denmark, Latvia, Netherlands, Poland, and deer isolates from the USA, had sufficient temporal signal in the data to reliably calibrate a time tree ($r^2 = 0.28-0.93$). Tip-calibration of another phylogeny comprising mink, deer and human background 1 isolates from these countries estimated the time to most recent common ancestor (tMRCA) to the 14th December 2019 (90% maximum posterior interval: 28th October 2019-8th February 2020), and the substitution rate to be $6.45 \pm 0.4$ s.d. $\times10^{-4}$ substitutions/site/year. These estimates are highly consistent with previous estimates[3], suggesting that our reconstructed time-scaled phylogenies are reliable. To determine host-specific rate variation, we extracted the terminal branch lengths of isolates corresponding to each host from this animal-human phylogeny (Fig. 5a). The distributions of terminal branch lengths was similar between the three hosts, with the substitution rate of SARS-CoV-2 in humans significantly exceeding that in minks (Mann–Whitney $U = 90690$, $p = 0.0120$). We performed the same analysis on reconstructed animal-human time trees for each country separately (Fig. 5b). With the exception of Latvia, no significant host-specific substitution rate variation was observed ($p > 0.05$).

**Predicted impact of candidate host-adaptive mutations on viral proteins.** We attempted to bioinformatically assess the impact of non-synonymous candidate mutations on protein function using *PROVEAN* scores[42] and their putative impact on viral fitness in a novel host using structural analyses. *PROVEAN* scores have been shown to correlate with how deleterious a mutation is to protein

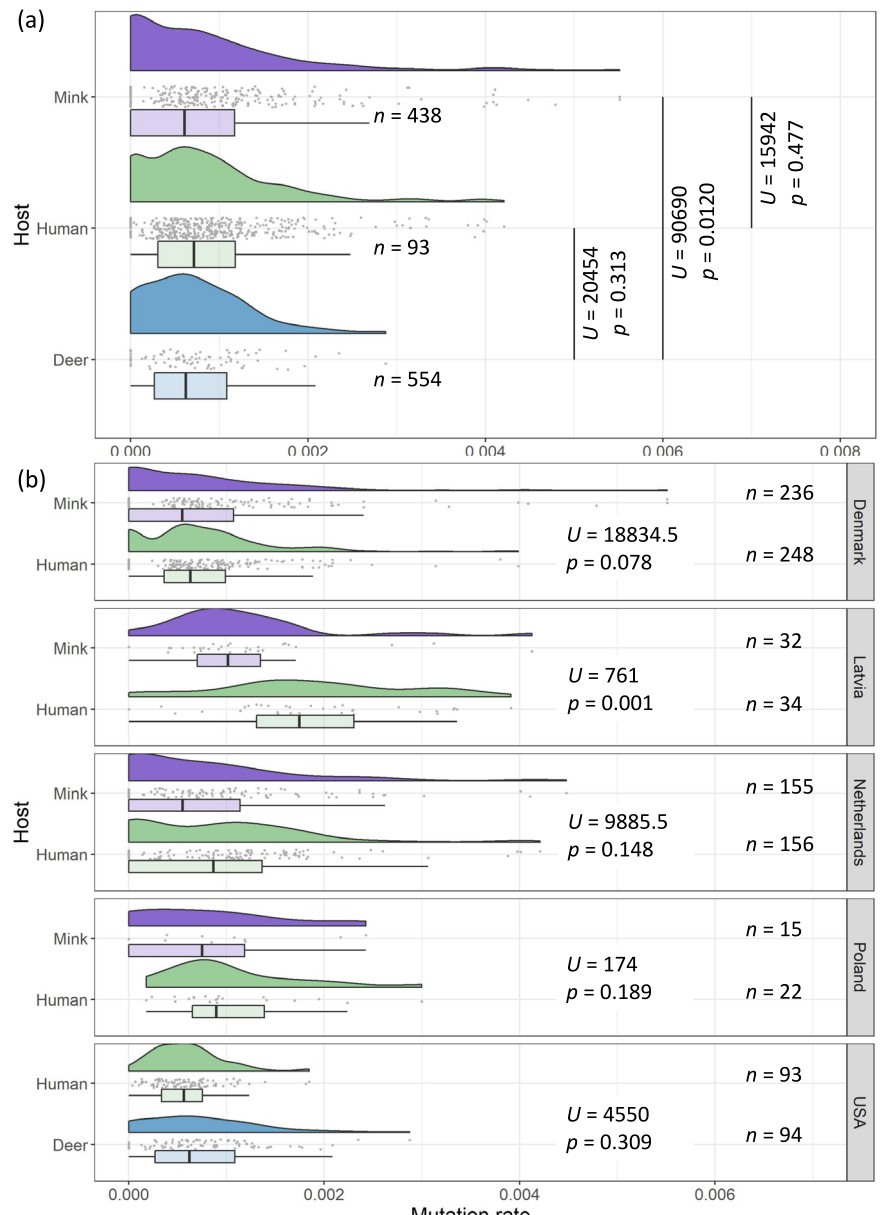

**Fig. 5 Host-specific substitution rate variation.** Raincloud plots[95] of terminal branch lengths stratified by **a** host, and **b** by both host and country. These plots comprise Gaussian kernel probability density, scatter and box-and-whisker plots (centre line, median; box limits, upper and lower quartiles; whiskers, 1.5x interquartile range). Multiple mink-human maximum-likelihood phylogenies of mink and human background 1 isolates were reconstructed and used for tip-calibration. Isolates that did not have complete dates or that were duplicate sequences were removed prior to analysis. The final number of isolates in each stratum that were used for tip-calibration, Mann–Whitney $U$-statistics and their associated $p$-values (based on a two-sided test), are annotated.

function[42]. They are computed based on the BLOSUM62 substitution matrix[43] whose substitution scores loosely reflect how biochemically conservative a mutation is[44], with positive scores implying more conservative mutations. Interestingly, most of the strong candidate mutations analysed are predicted to be conservative and functionally neutral (Table 2), including all mink-associated candidate mutations in the Spike protein. This is also the case for human-adaptive mutations implicated in immune escape and that have emerged recurrently in more transmissible viral lineages[18], suggesting that adaptive mutations, at least in the Spike protein, may not necessarily strongly impact protein function as assessed by these metrics.

Further, since N501T, F486L and Y453F fall within the SARS-CoV-2 Spike RBD, we considered their role in receptor binding affinity as putative sites of adaptation to a mink host. The Spike RBD (codon positions 319–541[45]) provides a critical region for

SARS-CoV-2 to attach to host cells via docking to ACE2 receptors, thereby allowing subsequent SARS-CoV-2 entry into host cells and eventual replication[46,47]. Specific residues within the RBD have been identified as critical for receptor binding[12,48,49], with potential to modulate both infectivity and antigenicity[45]. All three candidate Spike mutations (Y453F, F486L and N501T) identified suggest by our analyses are in residues directly involved in contacts in the Spike:ACE2 interface and are therefore relevant to the binding affinity and stability of the complex (Fig. S2).

We modelled various structures of mink or human ACE2 bound to the wild-type (WT, i.e., Wuhan-Hu-1 reference sequence) Spike protein, or to mutant Spike proteins carrying either N501T, F486L or Y453F. We then used the protein docking prediction protocols *HADDOCK*[50] and *mCSM-PPI2*[51] to analyse

**Table 2 PROVEAN scores of strong candidate mutations and other mutations previously described as adaptive to human and/or non-human hosts.**

| Protein | Mutation | Potentially adaptive to | Reference | PROVEAN score |
|---------|----------|------------------------|-----------|---------------|
| ORF1ab | NSP9_G37E | mink | present study | −5.108* |
|         | NSP3_L1035F | deer | | −0.808 |
| Spike | N501T | mink | [54], present study | 0.746 |
|       | F486L | | [55,54], present study | −0.035 |
|       | Y453F | | [29,54,30], present study | −0.393 |
|       | N501Y | Rodents, Humans | [18,94] | −0.090 |
|       | L452R | Humans | [18] | 0.559 |
|       | E484K | | | 0.128 |
|       | D614G | | | 0.598 |
|       | P681R | | | 0.741 |
| ORF3a | L219V | mink | present study | 0.276 |

Mutations that were predicted by PROVEAN to be deleterious to protein function are indicated by an asterisk (*).

(a) **HADDOCK scores**

| Spike structure | Human ACE2 | Mink ACE2 |
|-----------------|------------|-----------|
| WT (Wuhan-Hu-1) | -137.0 | -152.5 |
| Y453F | -145.0 | -155.1 |
| F486L | -139.8 | -159.5 |
| N501T | -137.6 | -148.7 |

(b) **mCSM-PPI2 (ΔΔG)**

| Substitution performed | Human ACE2 | Mink ACE2 |
|------------------------|------------|-----------|
| Y453F | -0.083 | -0.496 |
| F486L | -0.936 | -0.504 |
| N501T | 0.882 | 1.223 |

XXX – Destabilisation of complex
XXX – Stabilisation of complex
ΔΔG – Predicted change in binding energy

**Fig. 6 Predicted effects of candidate mutations. a** HADDOCK scores for the Spike:ACE2 complexes. More negative values relative to the WT-Spike:ACE2 complexes (highlighted in grey) indicate stronger binding energy of the complex. **b** mCSM-PPI2 predicted changes in binding energy (ΔΔG). Negative ΔΔG values are associated with destabilisation of the complex following mutation of the residue and positive values with stabilisation of the complex. Values in blue and red indicate predicted increases or decreases in complex stability respectively.

the change in stability of the Spike:ACE2 complexes due to each of these mutations (see "Methods"). We used this approach as previous work showed that it gave results that correlated well with experimental data on susceptibility to infection[12,52]. Interestingly, the stability predictions of both methods are somewhat conflicting, and indicate marginal changes in the stability of the complex (Fig. 6). Further, candidate mutations that are predicted to stabilise (or destabilise) the Spike:human-ACE2 complex are also predicted to stabilise (or destabilise) the Spike:mink-ACE2 complex (Fig. 6b). Overall, the PROVEAN and protein docking analyses are consistent with the hypothesis that SARS-CoV-2 mutations tend to be conservative and any small changes to structure caused by the candidate Spike mutations do not significantly affect the stability of the Spike:ACE2 complex.

## Discussion

Coronaviruses have placed an enormous burden on public health globally in recent years, including four endemic (human coronavirus HKU1, OC43, 229E and NL63), two epidemic (SARS, MERS), and most recently one pandemic species (SARS-CoV-2).

There is no doubt that novel coronaviruses will continue to emerge in humans. Therefore, understanding the cross-species transmission of SARS-CoV-2 and associated host adaptation is highly relevant to outbreak mitigation and future prevention. In this work, we analysed published and publicly available SARS-CoV-2 sequences isolated from animals compared with carefully selected human-associated sequences to understand the evolutionary events surrounding a host jump event.

Secondary host jumps of SARS-CoV-2 into animals have been documented for a variety of species, including cats and dogs, tigers and lions in zoos, farmed mink and wild deer in the USA. While in all cases, host range expansion arose through multiple independent spillover events, only those in mink and deer have led to the observation of extensive subsequent animal-to-animal transmission to date. Irrespective of the transmissibility potential of SARS-CoV-2, in different hosts, this is most likely due to companion animals and zoo animals having limited contact with congeners. While mink and deer spillovers were identified early, it is likely that SARS-CoV-2 has already established itself in other animal reservoirs that are less well-documented. For example, a

recent study of wild mustelids found three wild martens (*Martes martes*) and two badgers (*Meles meles*) to be seropositive for SARS-CoV-2[53]. Given the virus' prevalence in the human population and its ability to infect a broad range of mammalian hosts, it may be surprising if the number of non-human reservoir species did not increase.

Our analysis of animal SARS-CoV-2 isolates points to differing patterns of onward transmission in different sampled animal systems. We focused on deer and mink associated viral lineages for which phylogenetic transmission clusters have been well sampled and documented. Our analyses, focusing on a set of criteria applied to recurrent mutations, identify putative signatures of host adaptation following onward transmission of SARS-CoV-2 in mink and deer. The spike mutations N501T, F486L and Y453F have been shown to improve entry into cells expressing ferret ACE2 and are therefore animal adaptive[54]. Further, phylodynamic analyses of Dutch mink farm outbreaks have previously shown that viruses in minks that carry the Spike_F486L mutation may evolve and transmit at a faster rate[55]. Meanwhile, our functional prediction analyses using bioinformatic approaches suggest a minimal impact of all strong candidate Spike mutations, including Spike_F486L, on Spike: mink-ACE2 interactions. This is despite in vitro evidence that these three mutations allow more efficient cellular entry into cells expressing mustelid ACE2[54]. Further, while the strong candidate spike mutations Y453F and N501T were found to improve Spike:human-ACE2 interactions[56], we find that they confer minimal or no evolutionary advantage for transmission in humans, concordant with in vitro evidence that Y453F attenuates SARS-CoV-2 in human bronchial cells[54]. Together, these conflicting findings highlight the complex relationships between mutations and viral fitness. Additionally, the absence of strong candidate deer-adaptive mutations in the Spike protein, together with the presence of strong candidates in ORF1ab and ORF3a highlight the likely importance of mutations in non-Spike proteins, which remain poorly characterised. Further experimental investigations, particularly on the relationships between mutations and viral fitness, are warranted.

White-tailed deer present the best animal models for understanding the natural transmission of SARS-CoV-2 and constitute the first known animal reservoir of the virus, with locally high prevalence as documented by seropositivity of 30–40%[32,33,35]. Moreover, white-tailed deer populations are large, interconnected and distributed over a wide geographic range, including most of North America, Central America and parts of South America. Given the difficulties encountered by most worldwide governments to control the transmission of SARS-CoV-2 in humans, any attempt to eradicate the virus in white-tailed deer would be highly challenging, if even possible.

The culling of farmed minks in Denmark in late 2020, and the more recent speculation that Omicron might have evolved in rodents[57], highlight ongoing concerns over the emergence and accumulation of mutations while circulating in novel animal hosts following human-to-animal spillover, subsequently leading to the back-jump of more transmissible viral lineages into humans. Our results indicate that the putatively animal-adaptive mutations, for instance in mink lineages, likely confer minimal or no evolutionary advantage in humans, and as a result have remained at low frequencies. Additionally, our work suggests that the mutations accumulated while circulating in minks and deer have not caused drastic changes to the genomic landscape of SARS-CoV-2, since the relative proportions of nucleotide-nucleotide transitions occurring and the genomic composition in animal isolates largely mirror those in humans. Instead, we find a similar overrepresentation of C → U mutations in both human and animal hosts. Additionally, the most abundant

transitions after C → U are G → U, A → G and G → A. Some of these substitutions are consistent with systematic mutational pressures exerted by host-editing processes, involving APOBEC and ADAR proteins, and reactive oxygen species (C → U, A → G, and G → U, respectively)[58]. Of note is the subtle depletion of A → G mutations in minks vis-à-vis humans, which may reflect the differing activity of host ADAR in these species, though this would need to be experimentally validated. Nevertheless, these findings hint at similar mutagenic pressures in humans, mink and deer, which greatly overshadow those of host adaptation.

The current minimal levels of host-specific adaptation in mink and deer are reminiscent of our previous work early in the first wave of the COVID-19 pandemic, which failed to identify mutations in SARS-CoV-2 associated with increased transmissibility in humans[14]. The emergence of more transmissible VoCs driving the subsequent pandemic waves, highlight the strong collective, likely epistatic, phenotypic effects of multiple mutations. As such, while our analyses have not identified analogical 'animal-VoCs', this does not preclude the potential for new, more transmissible lineages to emerge in animal reservoirs in the future.

We could not find any crucial, prerequisite mutations for the secondary spillover of SARS-CoV-2 into mink and deer and observed no inflation of the substitution rates relative to that in its primary human host. These findings confirm that not only does human SARS-CoV-2 have the ability to infect multiple host species (i.e., broad host range), but it is also well pre-adapted to circulation in mink and deer despite significant ongoing adaptation to humans. This reinforces previous suggestions of SARS-CoV-2 as a 'generalist' virus[5]. This 'generalist' property may stem, in part, from the use of ACE2 as the primary host receptor for viral entry since the sequence and structure of ACE2 is fairly conserved across a broad range of mammals[10,12]. Other host pathways exploited by viral proteins, which determine transmission efficiency, may similarly be conserved. However, further experimental work identifying such host-viral interactions needs to be done.

A virus circulating in its natural host continues to evolve, indefinitely so, largely due to the pressure exerted by its host's immunity. Though, a faster rate of evolution may be expected soon after a successful jump into a novel host. By the time of sampling, human-associated SARS-CoV-2 lineages are still adapting to their human hosts, and their rate of evolution might still be inflated relative to their long-term future quasi-equilibrium. As such, the fact that we did not observe a higher rate of evolution of viral lineages circulating in mink and deer at this stage, should not necessarily be interpreted as an absence of selective pressure in its novel animal hosts, but rather as a heightened selection on viruses circulating in humans not having yet relaxed.

We note several limitations of our present study. The phylogenetically distinct clusters that we manually curated do not necessarily correspond to discrete spillover events between an individual and a single animal. In fact, as demonstrated by the mink_Denmark_1 cluster, complex transmission patterns are difficult to disentangle solely based on sequence information alone. This is further exacerbated by the difficulty of identifying and sequencing every human or non-human host within any transmission chain[59]. Transmission chain reconstruction (i.e., 'who-infected-whom') using *SeqTrack*[59] or *TransPhylo*[60] may provide a more reliable estimate of the number of individual spillover events, but is beyond the scope of our study. Separately, SARS-CoV-2 surveillance in animals early on in the pandemic was minimal or absent so we cannot rule out the possibility that some early animal outbreaks were left undetected, and that some animal-specific mutations may have been introduced into the

global diversity of SARS-CoV-2 circulating in humans during this period. As such, our claim that the emergence of animal-adaptive mutations in humans largely predates human-to-animal transmission is restricted to documented spillover events. Additionally, our approach to identify putatively adaptive alleles may not be able to detect these animal-specific mutations as it relies on a comparison of animal-associated allele frequencies against that from a human background. For our bioinformatic functional analyses, the performance of PROVEAN on assessing the functional impact of mutations has not been specifically validated on viral sequence datasets, so it remains unclear whether the default score threshold can be used to reliably identify putatively 'deleterious' mutations. Additionally, while our PROVEAN and structural analyses attempt to assess the effects of mutations on protein structure and function, it is difficult to interpret whether these effects (or lack thereof) directly affect fitness and the mechanisms for doing so. Mutational studies in vitro or in vivo are key in elucidating such mechanisms and may shed light on the broader strategies that SARS-CoV-2 employ to adapt for circulation in novel host species.

Overall, our findings indicate that the mutational prerequisite for efficient SARS-CoV-2 transmission in novel hosts is low, highlighting the 'generalist' nature of SARS-CoV-2 as a mammalian pathogen. In light of this, human-to-animal and spillback events are both a realised and likely outcome of widespread SARS-CoV-2 transmission in human populations. The establishment of SARS-CoV-2 in animal reservoirs further challenges the adoption of a suppression/elimination strategy to pandemic mitigation since back-spill to human populations, as seen in association with Danish and Dutch mink farms, seem to be inevitable. Our results indicate that putatively animal-adaptive mutations have emerged in the short time that SARS-CoV-2 was circulating in mink and deer, but that these mutations do not appear to confer a significant advantage for circulation in humans. Nevertheless, mutational surveillance of SARS-CoV-2 in human and animal populations remains important to document the adaptive potential of the virus and its consequences in human and animal hosts.

## Methods

**Data acquisition**. All animal SARS-CoV-2 isolates that were present in the 17 March 2022 release of the Audacity (UShER[61]) tree on GISAID[19,20] were retrieved (Table 1). Additionally, human accessions were subsampled from the Audacity tree based on various inclusion criteria depending on the analysis performed. The inclusion criteria used for each analysis are described in the 'Human backgrounds' section. The alignments of human and animal genomes (to WIV04; EPI_ISL_402124) corresponding to these accessions were then extracted from the masked multiple sequence alignment (26th March 2022) on GISAID using the subseq utility of Seqtk (https://github.com/lh3/seqtk).

**Maximum likelihood and Audacity phylogenies**. Maximum-likelihood trees were inferred from the masked genomic sequence alignments using IQ-Tree2[62], specifying a GTR + Γ substitution model. All trees were either visualised using Dendroscope 3[63] or ggtree[64], and manipulated using the Ape package[65] in R. Where the number of isolates considered is large, we extracted subtrees from the Audacity tree for further analysis using the drop.tip function in the R package, Ape v5.5[65]. This was to avoid the excessive computational overhead of phylogenetic reconstruction.

**Animal SARS-CoV-2 cluster annotation**. To place animal SARS-CoV-2 isolates in the context of human infections, we visualised a subsampled Audacity tree, representing the global genomic diversity of SARS-CoV-2 (Fig. 1a). A total of 16,911 isolates, comprising ten human SARS-CoV-2 isolates per country per lineage, in addition to all isolates shown in Table 1, were included in this subsampled tree. Separately, we visually inspected a subsampled Audacity tree comprising animal isolates and all human isolates collected prior to the most recent animal isolates in each country (mink: 1,201,639 isolates; deer: 1,698,656 isolates). The accessions considered in these analyses are provided in Supplementary Data 2–4, respectively. This was to identify phylogenetically distinct clusters of animal isolates representing independent spillover events. Monophyletic clades of animal SARS-CoV-2 isolates that were assigned the same PANGO lineage[40] were initially designated as separate clusters. These preliminary clusters were manually

inspected, and subsequently merged or separated based on their phylogenetic placement. In addition, we reconstructed mink or deer-only phylogenies ($n = 928$ and 95, respectively) rooted to GISAID reference genome WIV04 using ultrafast bootstrapping (UFBoot)[66] and approximate likelihood-ratio tests (SH-aLRT)[67] with 1000 replicates. The final identified clusters were monophyletic clades supported by ≥90% SH-aLRT and ≥93% UFBoot branch support scores. These mink and deer-only phylogenies annotated with branch support scores are shown in Supplementary Data 5 and 6. Cluster information of all animal accessions included in this study is provided in Supplementary Data 7.

**Identifying recurring mutations**. The maximum-likelihood trees and corresponding alignments of SARS-CoV-2 isolates associated with a single host species (i.e., mink or deer) were screened for homoplasies using HomoplasyFinder v0.0.0.9[68]. Homoplasies are mutations that have emerged recurrently and independently throughout a taxon's evolutionary history and may be indicative of host adaptation. HomoplasyFinder employs the method first described by Fitch[69], providing, for each site, the site specific consistency index and the minimum number of independent emergences in the phylogenetic tree. All nucleotide positions with a consistency index <0.5 are considered homoplastic.

**Human backgrounds**. In our analyses, we compared mink or deer-associated SARS-CoV-2 isolates to different subsamples of human isolates. Selection of appropriate human backgrounds to identify patterns of host-specific adaptation is crucial to minimise the risk of artefactual results. Depending on the inclusion criteria of human isolates, the inferences that can be made differ greatly. In this study, the main human background (referred to as 'human background 1') comprises human isolates with countries of isolation, PANGO lineages, and range of sampling dates matching those for animal isolates (±1 month). Additionally, human isolates that fulfilled these criteria were randomly subsampled to match the number of viral isolates per PANGO lineage in animals (where possible). This human background controls for biases in the relative sizes of SARS-CoV-2 lineages, genomic diversity, and sequencing effort. A second human background (referred to as 'background 2') comprising 10 human isolates for each PANGO lineage present within the countries of isolation, regardless of sampling date, was also used. This background allows us to compare animal-specific vis-à-vis human-specific adaptation of SARS-CoV-2 in a wider evolutionary context.

**Allele frequency and mutational biases**. Allele frequencies and nucleotide-nucleotide transitions (e.g. number of $C \rightarrow U$ mutations) were computed for all positions in the animal or human SARS-CoV-2 masked sequence alignment using the base.freq function from the Ape package in custom R scripts. We tested whether the frequency of nucleotide-nucleotide transitions in human and animal genomes differed using a Monte Carlo simulation of the $\chi^2$ statistic with fixed margins (2000 iterations)[70,71]. This was implemented using the chisq.test function in R with the simulate.p.value flag. Dinucleotide frequencies were computed using the dinucleotideFrequency in the Biostrings[72] package in R. A permutation test for 1000 iterations was performed to determine if the average number of $A \rightarrow G$ transitions differed between human and mink-associated isolates. Briefly, for each iteration, we randomised the host labels of mink and human SARS-CoV-2 isolates and computed the change in log10-transformed ratio of the proportion of $A \rightarrow G$ transitions in animal to that for human isolates. We then calculated the p-value as the proportion of iterations where the computed metric was lesser than that observed without permutation. Separately, ordination of host-specific dinucleotide frequencies was performed via a principal components analysis with the prcomp function in R. Dinucleotide frequencies were zero-centred and scaled to unit variance prior to ordination. The accessions used for these analyses are provided in Supplementary Data 8–11.

**Estimating host-specific substitution rates**. Animal isolates, stratified by country, were analysed relative to human isolates from the same country and isolation timespan. Phylogenies of human and animal SARS-CoV-2 isolates were informally assessed for temporal signal via linear regression of root-to-tip distances against time, using TreeTime[73]. These phylogenies were then tip-calibrated using TreeTime under an uncorrelated relaxed clock model, with a normal prior on rate heterogeneity across branches. Additionally, tip-calibration was run using a Kingman coalescent tree prior with an effective population size estimated using a skyline[74]. The terminal branch lengths of the inferred divergence trees were divided by those of the time-scaled trees to obtain estimates of the host-specific mean substitution rates in substitutions per site per year. We tested if the distributions of terminal branch lengths differed between hosts by performing two-sided Mann–Whitney U-tests using the wilcox.test function in R. Isolates with ambiguous sampling dates were excluded from this analysis. Identical sequences were randomly removed using the rmdup utility of SeqKit[75]. The final mink and deer accessions used in these substitution rate analyses are provided in Supplementary Data 12 and 13, respectively.

**Predicting changes in the stability of viral proteins following mutation**. We used the PROVEAN web server[76] to bioinformatically assess the functional impact of candidate adaptive mutations on viral proteins. The PROVEAN score is an

alignment-based metric that determines the change in sequence similarity of a protein given a single amino acid substitution, which was shown to correlate well with the functional impact of that mutation[42]. *PROVEAN* scores that are ≤−2.5 are classified as 'deleterious' mutations.

Additionally, we modelled various versions of the Spike:ACE2 complex to determine the change in stability of the Spike:ACE2 complex due to mutation. The structure of the wild-type (WT; i.e., Wuhan-Hu-1 reference sequence) SARS-CoV-2 Spike protein bound to human ACE2 has been solved at 2.45Å resolution[77] (Protein Data Bank[78] (PDB) ID 6M0J). We visualised this structure using *PyMOL v2.4.1*[79]. We used this as the template to model various structures of ACE2 bound to the SARS-CoV-2 Spike protein. In particular, we modelled structures of mink-ACE2 bound to the WT-Spike protein, and human- or mink-ACE2 bound to mutant Spike proteins carrying either of the candidate mutations Y453F, F486L or N501T. We generated query–template alignments using *HH-suite*[80] and predicted 3D models using *MODELLER v.9.24*[81]. We used the 'very_slow' schedule for model refinement to optimise the geometry of the complex and interface. We generated 10 models for each Spike:ACE2 complex and selected the model with the lowest nDOPE[82] score, which reflects the quality of the model. Positive scores are likely to be poor models, while scores >−1 likely to be native-like. The sequence similarity of the human ACE2 and the mink ACE2 is fairly high (83% amino acid sequence identity), and all generated models were of high quality (nDOPE < −1).

Following successful modelling of the various Spike:ACE2 complexes, two independent methods were used to assess changes to complex stability. The first, *HADDOCK*[51], is one of the top-performing protein-protein docking servers in the CAPRI competition[83]. The *HADDOCK* score is a weighted sum of various predicted energy values (i.e., van der Waals, electrostatics and desolvation). We used the *HADDOCK v2.4* webserver to score all complexes (Fig. 6a). We then compared the scores of WT-Spike:human/mink-ACE2 to mutant-Spike:human/mink-ACE2 complexes. We also calculated the predicted change in binding energy (ΔΔG) of the Spike:ACE2 complexes using *mCSM-PPI2*[52] (Fig. 6b). This programme assigns a graph-based signature vector to each mutation, which is then used within machine learning models to predict the change in binding energy following an amino acid substitution. The signature vector is based upon atom-distance patterns in the protein, pharmacophore information and available experimental information, evolutionary information, and energetic terms. We used the *mCSM-PPI2* server (http://biosig.unimelb.edu.au/mcsm_ppi2/) for the simulations. In particular, we simulated the mutation of the WT-Spike (i.e., Y453F, F486L or N501T) while bound to human or mink-ACE2. For *HADDOCK*, a more negative value than for the reference WT-Spike:ACE2 complex suggests stabilisation of the complex. Meanwhile, for *mCSM-PPI-2*, negative and positive ΔΔG values reflect destabilisation and stabilisation of the complex by the mutation, respectively. These two methods were used because we found in a previous study that the reported stability changes following mutations in the Spike:ACE2 complex correlated well with the available in vivo and in vitro experimental data on susceptibility to infection[12].

**Reporting summary**. Further information on research design is available in the Nature Research Reporting Summary linked to this article.

## Data availability
All genomic sequences used in this study are publicly available on registration at GISAID (https://www.gisaid.org/). The accessions for all sequences analysed are listed in Supplementary Data 2–4, 8–13. All raw data files required to reproduce the analyses in this paper can be downloaded from Zenodo (https://doi.org/10.5281/zenodo.6528187). The structure of the wild-type SARS-CoV-2-Spike:human-ACE2 can be downloaded from the PDB under the accession 6M0J (https://www.rcsb.org/structure/6m0j).

## Code availability
All custom code used to perform the analyses are hosted on GitHub (https://github.com/cednotsed/ditto.git)[84]. For all nucleotide transitions, the corresponding amino acid residue positions and changes were determined using an association table generated using a custom *Python* 3.7.11 script hosted on GitHub (https://github.com/cednotsed/SARS-CoV-2-hookup).

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

## Acknowledgements

We would like to acknowledge and thank all the originating and submitting laboratories of the SARS-CoV-2 sequences deposited on GISAID on which our analyses are based. This work is supported by the NIHR Precision AMR award received by D.R., UCL Excellence Fellowship received by L.v.D and the BBSRC equipment grant (BB/R01356X/1) received by F.B.

## Author contributions

C.C.S.T., L.v.D. and F.B. designed the study. C.C.S.T. and S.D.L. performed the computational analyses; C.C.S.T., L.v.D. and F.B. wrote the paper with inputs from all co-authors. D.R., C.J.O, D.B., C.O., M.S.N., S.V.K. and V.K. provided feedback, contributed to manuscript editing, literature review, and interpretation of the results.

## Competing interests

The authors declare no competing interests.
