## [Peer Review File · Nature Communications]

Transmission of SARS-CoV-2 from humans to animals and potential host adaptationReviewers' Comments:

Reviewer #1:

Remarks to the Author:

SUMMARY:

This is an interesting study that looks at recurrent potentially adaptive mutations that occur after SARS-CoV-2 undergoes reverse zoonoses from humans to other species. The main finding is that these reverse zoonoses are not associated with a large increase in evolutionary rate. For mink there are a few putatively adaptive mutations, less so for deer.

Overall, the methods used for the phylogenetic assignments of independent animal outbreaks and the identification of putatively adaptive mutations are a bit ad hoc and manual. This is probably OK for the latter, but I suggest a bit more rigor or at least explanation for the former (see my major comment). I also have a few minor comments.

But in total, this is a good and interesting paper that should be suitable for publication after minor revisions.

MAJOR COMMENTS:

- The number of unique outbreaks in animal species appear to be established by manual inspection of the global phylogenetic tree. I am not necessarily disagreeing with the conclusions, but it's hard to establish they are correct. On the SARS-CoV-2 trees, the support for particular topologies is often low due to closely related sequences, so it's difficult to assess from the trees in Figure 1 (which don't even have branch lengths shown) how well supported the monophyletic or paraphyletic nature of the clades are. Ideally this could be assessed statistically, or if done manually at least some more thought and explanation needs to be put into this point.

MINOR COMMENTS:

- Lines 53-54: although it is generally recognized SARS-CoV-2 is a bat coronavirus, it remains unclear whether the initial infection occurred due to zoonotic spillover or a lab accident.

- Lines 73-74: The R_0 of SARS-CoV-2 is thought to have increased substantially from early 2020 to the present due to adaptation to humans. The description of the R_0 should be re-worded to reflect this fact.

- Lines 75-77: There is evidence consistent with the idea that SARS-CoV-2 is probably derived from ancestral viruses that can bind efficiently to human ACE2. However, there is no evidence that the ancestral bat associated lineages could undergo efficient human-to-human transmission. This is possible but not known, and none of the cited references support this point.

- Deep mutational scanning data (Starr et al, Cell, 2020) indicate that the mink adaptation mutations Y453F and N501T both increase affinity to human ACE2 as well. This may be worth mentioning, as it seems these direct experimental measurements are preferable to the molecular modeling used in the paper.

Reviewer #2:

Remarks to the Author:

Tan et al. have used sequences of SARS-CoV-2 from human, mink, and deer to identify non-human

specific adaptive mutations, and investigate and compare SARS-CoV-2 mutation load in these species. This work gives an insight into the evolution of SARS-CoV-2 in humans, minks and deer showing that SARS-CoV-2 viruses are still evolving in these species, and, to date, no mutation found in mink/deer variants seem to confer advantage for spillback to humans.

I believe the work was well designed and found the manuscript very enjoyable to read.

I do have a few questions/suggestions that I hope the authors will be able to address.

(1) Page 10, line 249. "Since we find no evidence of strong positive selection [...] carriage of these alleles across many countries is not expected". Would the authors think that the widespread presence of these (unexpectedly widespread) mutations associated with the presence of other mutations that confer advantage in humans? Are these associated with a specific human variant that was more transmissible?

(2) Page 11, Fig. 3b and c. I would suggest the indication of the time of the first appearance of each candidate mutation (e.g., as a vertical line) to be marked on the panels, so it is easier for the readers to see which appeared before or after in humans.

(3) Page 15, line 345. Are the candidate mutations ever found in the same viral sequence? Could it be that individually they do not confer binding stability but (especially for the mutation of the Spike protein) that they are physically close enough that they do when occurring in combination? Is it possible to do the modelling with more than one substitution at a time?

We would like to thank the editor, Dr. Emily Jones, and the reviewers for taking the time to assess our manuscript for consideration in *Nature Communications*. In our revised manuscript, we performed all our analyses again on the latest 17th March Audacity release on GISAID, which includes an additional 139 mink and 22 deer associated SARS-CoV-2 genomes. Additionally, we used a different approach to identify independent phylogenetically distinct animal clusters as per Reviewer 1's suggestion. Following this reanalysis, the main results and conclusions of the study remain unchanged, highlighting the robustness of our findings. Please find our point-by-point response inline.

REVIEWER COMMENTS

Reviewer #1 (Remarks to the Author):

SUMMARY:

This is an interesting study that looks at recurrent potentially adaptive mutations that occur after SARS-CoV-2 undergoes reverse zoonoses from humans to other species. The main finding is that these reverse zoonoses are not associated with a large increase in evolutionary rate. For mink there are a few putatively adaptive mutations, less so for deer.

Overall, the methods used for the phylogenetic assignments of independent animal outbreaks and the identification of putatively adaptive mutations are a bit ad hoc and manual. This is probably OK for the latter, but I suggest a bit more rigor or at least explanation for the former (see my major comment). I also have a few minor comments.

But in total, this is a good and interesting paper that should be suitable for publication after minor revisions.

Thank you for your positive assessment.

MAJOR COMMENTS:

- The number of unique outbreaks in animal species appear to be established by manual inspection of the global phylogenetic tree. I am not necessarily disagreeing with the conclusions, but it's hard to establish they are correct. On the SARS-CoV-2 trees, the support for particular topologies is often low due to closely related sequences, so it's difficult to assess from the trees in Figure 1 (which don't even have branch lengths shown) how well supported the monophyletic or paraphyletic nature of the clades are. Ideally this could be assessed statistically, or if done manually at least some more thought and explanation needs to be put into this point.

We completely agree that the low genetic diversity and hence lack of support SARS-CoV-2 phylogenies is an area of concern. Therefore, to assess the confidence in our cluster assignments, we performed approximate likelihood-ratio tests (SH-aLRT) and ultrafast-bootstrapping approach (UFboot) on the reconstructed mink- and deer-only phylogenies. The final cluster assignments were represented as well-supported monophyletic clades with support values of

$\geq 89.9\%$ SH-aLRT AND $\geq 93\%$ UFBoot. We have provided a description of this analysis in the methods section. Additionally, these bootstrapped phylogenies can be found in Fig. S3.

MINOR COMMENTS:

- Lines 53-54: although it is generally recognized SARS-CoV-2 is a bat coronavirus, it remains unclear whether the initial infection occurred due to zoonotic spillover or a lab accident.

We agree with you and have qualified the sentence to reflect this uncertainty.

- Lines 73-74: The R_0 of SARS-CoV-2 is thought to have increased substantially from early 2020 to the present due to adaptation to humans. The description of the R_0 should be re-worded to reflect this fact.

We have now included the notion of increasing R_0 values in lines 82-83.

- Lines 75-77: There is evidence consistent with the idea that SARS-CoV-2 is probably derived from ancestral viruses that can bind efficiently to human ACE2. However, there is no evidence that the ancestral bat associated lineages could undergo efficient human-to-human transmission. This is possible but not known, and none of the cited references support this point.

We have now acknowledged the lack of experimental evidence in lines 77-78.

- Deep mutational scanning data (Starr et al, Cell, 2020) indicate that the mink adaptation mutations Y453F and N501T both increase affinity to human ACE2 as well. This may be worth mentioning, as it seems these direct experimental measurements are preferable to the molecular modeling used in the paper.

We have included mention of the findings of this study in lines 421-422.

Reviewer #2 (Remarks to the Author):

Tan et al. have used sequences of SARS-CoV-2 from human, mink, and deer to identify non-human specific adaptive mutations, and investigate and compare SARS-CoV-2 mutation load in these species. This work gives an insight into the evolution of SARS-CoV-2 in humans, minks and deer showing that SARS-CoV-2 viruses are still evolving in these species, and, to date, no mutation found in mink/deer variants seem to confer advantage for spillback to humans. I believe the work was well designed and found the manuscript very enjoyable to read. I do have a few questions/suggestions that I hope the authors will be able to address.

Thank you for your positive comments.

(1) Page 10, line 249. “Since we find no evidence of strong positive selection [...] carriage of these alleles across many countries is not expected”. Would the authors think that the widespread presence of these (unexpectedly widespread) mutations associated with the presence of other

mutations that confer advantage in humans? Are these associated with a specific human variant that was more transmissible?

I82T and D377Y are indeed associated with the Delta variant that was prevalent in the human population before the emergence of the Omicron variant, and this may explain their presence in virus isolated from multiple countries. However, after some thought, we consider it preferable to only highlight the six strongest mink adapted candidates in Figure 3. We have also revised lines 243-254 to this end.

(2) Page 11, Fig. 3b and c. I would suggest the indication of the time of the first appearance of each candidate mutation (e.g., as a vertical line) to be marked on the panels, so it is easier for the readers to see which appeared before or after in humans.

We have now included a red dashed line indicating the beginning of the mink outbreaks in Netherlands or the deer outbreaks in the USA (Fig. 3b and 3c) so it is easier to see how many human isolates possessed the candidate mutations prior to the first detected anthroponotic event.

(3) Page 15, line 345. Are the candidate mutations ever found in the same viral sequence? Could it be that individually they do not confer binding stability but (especially for the mutation of the Spike protein) that they are physically close enough that they do when occurring in combination? Is it possible to do the modelling with more than one substitution at a time?

Of the 928 mink sequences we have in our dataset, only 14 isolates contain at least two of the spike mutations (Y453F, F486L, N501T), of which all 14 possess F486L and N501T. We agree that epistasis is plausible, however the cooccurrence of these spike mutations is relatively rare.

The largely inconclusive modelling results in our study suggest that either the effects of these spike mutations on ACE2 binding/cellular entry are complex and/or that our structural analyses lack sufficient sensitivity to resolve the impact of interactions involving these mutations. We maintain that *in vitro* studies are necessary to probe the full effects and have noted this in the limitations paragraph of the discussion (lines 506-511).

Reviewers' Comments:

Reviewer #1:

Remarks to the Author:

I support publication of the revised manuscript.

Reviewer #2:

Remarks to the Author:

I am satisfied with the authors' revisions.